# A Three-Dimensional ResNet and Transformer-Based Approach to Anomaly Detection in Multivariate Temporal–Spatial Data

**DOI:** 10.3390/e25020180

**Published:** 2023-01-17

**Authors:** Lijuan Xu, Xiao Ding, Dawei Zhao, Alex X. Liu, Zhen Zhang

**Affiliations:** 1Shandong Provincial Key Laboratory of Computer Networks, Shandong Computer Science Center (National Supercomputer Center in Jinan), Qilu University of Technology (Shandong Academy of Sciences), Jinan 250014, China; 2Computer Science and Technology, Harbin Institute of Technology, Weihai 264209, China; 3Technology Research Institute of Cyberspace Security of Harbin Institute, Harbin 150001, China

**Keywords:** anomaly detection, multivariate temporal–spatial data, deep learning

## Abstract

Anomaly detection in multivariate time series is an important problem with applications in several domains. However, the key limitation of the approaches that have been proposed so far lies in the lack of a highly parallel model that can fuse temporal and spatial features. In this paper, we propose TDRT, a three-dimensional ResNet and transformer-based anomaly detection method. TDRT can automatically learn the multi-dimensional features of temporal–spatial data to improve the accuracy of anomaly detection. Using the TDRT method, we were able to obtain temporal–spatial correlations from multi-dimensional industrial control temporal–spatial data and quickly mine long-term dependencies. We compared the performance of five state-of-the-art algorithms on three datasets (SWaT, WADI, and BATADAL). TDRT achieves an average anomaly detection F1 score higher than 0.98 and a recall of 0.98, significantly outperforming five state-of-the-art anomaly detection methods.

## 1. Introduction

### 1.1. Motivation

Anomaly detection is the core technology that enables a wide variety of applications, such as video surveillance, industrial anomaly detection, fraud detection, and medical anomaly detection. Traditional approaches use clustering algorithms [1] and probabilistic methods [2]. However, clustering-based approaches have limitations, with the possibility of a dimensional disaster as the number of dimensions increases. Probabilistic-based approaches require a lot of domain knowledge. Recently, deep learning-based approaches, such as DeepLog [3], THOC [4], and USAD [5], have been applied to time series anomaly detection. DeepLog uses long short-term memory (LSTM) to learn the sequential relationships of time series. THOC uses a dilated recurrent neural network (RNN) to learn the temporal information of time series hierarchically. USAD combines generative adversarial networks (GAN) and autoencoders to model multidimensional time series. Their key advantages over traditional approaches are that they can mine the inherent nonlinear correlation hidden in large-scale multivariate time series and do not require artificial design features.

### 1.2. Limitations of Prior Art

The key limitation of this deep learning-based anomaly detection method is the lack of highly parallel models that can fuse the temporal and spatial features. As such, most of these approaches rely on the time correlation of time series data for detecting anomalies. The lack of such a model limits the further development of deep learning-based anomaly detection technology. Without such a model, it is difficult to achieve an anomaly detection method with high accuracy, a low false alarm rate, and a fast detection speed.

### 1.3. Proposed Approach

In this paper, we propose TDRT, a three-dimensional ResNet and transformer-based anomaly detection method. TDRT can automatically learn the multi-dimensional features of temporal–spatial data to improve the accuracy of anomaly detection. TDRT is composed of three parts. The first part is three-dimensional mapping of multivariate time series data, the second part is time series embedding, and the third part is attention learning. TDRT combines the representation learning power of a three-dimensional convolution network with the temporal modeling ability of a transformer model. Via the three-dimensional convolution network, our model aims to capture the temporal–spatial regularities of the temporal–spatial data, while the transformer module attempts to model the longer- term trend.

Our TDRT model advances the state of the art in deep learning-based anomaly detection on two fronts. First, it provides a method to capture the temporal–spatial features for industrial control temporal–spatial data. Our model shows that anomaly detection methods that consider temporal–spatial features have higher accuracy than methods that only consider temporal features. Second, our model has a faster detection rate than the approach that uses LSTM and one-dimensional convolution separately and then fuses the features because it has better parallelism.

### 1.4. Technical Challenges and Our Solutions

The first challenge is to obtain the temporal–spatial correlation from multi-dimensional industrial control temporal–spatial data. This is challenging because the data in an industrial system are affected by multiple factors. The value of a sensor or controller may change over time and with other values. For example, SWAT [6] consists of six stages from P1 to P6; pump P101 acts on the P1 stage, and, during the P3 stage, the liquid level of tank T301 is affected by pump P101. When the value of the pump in the P1 stage is maliciously changed, the liquid level of the tank in the P3 stage will also fluctuate. Hence, it is beneficial to detect abnormal behavior by mining the relationship between multidimensional time series. Therefore, we use a three-dimensional convolutional neural network (3D-CNN) to capture the features in two dimensions. The advantage of a 3D-CNN is that its cube convolution kernel can be convolved in the two dimensions of time and space.

The second challenge is to build a model for mining a long-term dependency relationship quickly. During a period of operation, the industrial control system operates in accordance with certain regular patterns. To address this challenge, we use the transformer to obtain long-term dependencies. The advantage of the transformer lies in two aspects. On the one hand, its self-attention mechanism can produce a more interpretable model, and the attention distribution can be checked from the model. On the other hand, it has less computational complexity and can reduce the running time.

### 1.5. Key Technical Novelty and Results

The key technical novelty of this paper is two fold. First, we propose a approach that simultaneously focuses on the order information of time series and the relationship between multiple dimensions of time series, which can extract temporal and spatial features at once instead of separately. Second, we propose a approach to apply an attention mechanism to three-dimensional convolutional neural network. We evaluated TDRT on three data sets (SWaT, WADI, BATADAL). Our results show that TDRT achieves an anomaly recognition precision rate of over 98% on the three data sets.

## 2. Industrial Control Network and Threat Model

### 2.1. Industrial Control Network

With the rapid development of the Industrial Internet, the Industrial Control Network has increasingly integrated network processes with physical components. The physical process is controlled by the computer and interacts with users through the computer. The local fieldbus communication between sensors, actuators, and programmable logic controllers (PLCs) in the Industrial Control Network can be realized through wired and wireless channels. Commands are sent between the PLC, sensors, and actuators through network protocols, such as industrial EtherNet/IP, common industrial protocol (CIP), or Modbus. The process control layer network is the core of the Industrial Control Network, including human–machine interfaces (HMIs), the historian, and a supervisory control and data acquisition (SCADA) workstation. The HMI is used to monitor the control process and can display the historical status information of the control process through the historical data server. The historian is used to collect and store data from the PLC. The role of the supervisory control and data acquisition (SCADA) workstation is to monitor and control the PLC.

### 2.2. Threat Model

The Industrial Control Network plays a key role in infrastructure (i.e., electricity, energy, petroleum, and chemical engineering), smart manufacturing, smart cities, and military manufacturing, making the Industrial Control Network an important target for attackers [7,8,9,10,11]. Adversaries have a variety of motivations, and the potential impacts include damage to industrial equipment, interruption of the production process, data disclosure, data loss, and financial damage. Factors such as insecure network communication protocols, insecure equipment, and insecure management systems may all become the reasons for an attacker’s successful intrusion. As shown in Figure 1, the adversary can attack the system in the following ways:Intruders can attack sensors, actuators, and controllers. For example, attackers modify the settings or configurations of sensors, actuators, and controllers, causing them to send incorrect information [12].Intruders can attack the HMI. For example, attackers exploit vulnerabilities in their software to affect the physical machines with which they interact.Intruders can attack the network. For example, attackers can affect the transmitted data by injecting false data, replaying old data, or discarding a portion of the data.Intruders can physically attack the Industrial Control Network components. For example, attackers can maliciously modify the location of devices, physically change device settings, install malware, or directly manipulate the sensors.

This paper considers a powerful adversary who can maliciously destroy the system through the above attacks. Their ultimate goal is to manipulate the normal operations of the plant. Attacks can exist anywhere in the system, and the adversary is able to eavesdrop on all exchanged sensor and command data, rewrite sensors or command values, and display false status information to the operators. Attackers attack the system in different ways, and all of them can eventually manifest as physical attacks. Therefore, we can detect anomalies by exploiting the deviation of the system caused by changes in the sensors and instructions.

## 3. Related Works

Anomaly detection is a challenging task that has been largely studied. Anomalies can be identified as outliers and time series anomalies, of which outlier detection has been largely studied [13,14,15,16]; however, this work focuses on the overall anomaly of multivariate time series. In industrial control systems, such as water treatment plants, a large number of sensors work together and generate a large amount of measurement data that can be used for detection. These measurement data restrict each other, during which a value identified as abnormal and outside the normal value range may cause its related value to change, but the passively changed value may not exceed the normal value range. Therefore, it is necessary to study the overall anomaly of multivariate time series within a period [17]. Among the different time series anomaly detection methods that have been proposed, the methods can be identified as clustering, probability-based, and deep learning-based methods.

### 3.1. Clustering-Based

Clustering-based anomaly detection methods leverage similarity measures to identify critical and normal states. The key to this approach lies in how to choose the similarity, such as the Euclidean distance and shape distance. Clustering methods initially use the Euclidean distance as a similarity measure to divide data into different clusters. Almalawi [1] proposed a method that applies the DBSCAN algorithm [18] to cluster supervisory control and data acquisition (SCADA) data into finite groups of dense clusters. Then, the critical states are sparsely distributed and have large anomaly scores. Because DBSCAN is not sensitive to the order of the samples, it is difficult to detect order anomalies. In addition, this method is only suitable for data with a uniform density distribution; it does not perform well on data with non-uniform density. Due to the particularity of time series, a k-shape clustering method for time series has been proposed [19], which is a shape distance-based method. Boniol et al. proposed a SAND algorithm by extending the k-shape algorithm, which is designed to adapt and learn changes in data features [20]. This is a technique that has been specifically designed for use in time series; however, it mainly focuses on temporal correlations and rarely on correlations between the dimensions of the time series.

### 3.2. Probability-Based

Anomaly detection has also been studied using probabilistic techniques [2,21,22,23,24]. The traditional hidden Markov model (HMM) is a common paradigm for probability-based anomaly detection. The idea is to estimate a sequence of hidden variables from a given sequence of observed variables and predict future observed variables. However, the HMM has the problems of a high false-positive rate and high time complexity. Chen and Chen alleviated this problem by integrating an incremental HMM (IHMM) and adaptive boosting (Adaboost) [2]. However, they only test univariate time series. Melnyk proposed a method for multivariate time series anomaly detection for aviation systems [23]. The approach models the data using a dynamic Bayesian network–semi-Markov switching vector autoregressive (SMS-VAR) model. However, it lacks the ability to model long-term sequences. Motivated by the problems in the above method, Xu [25] proposed an anomaly detection method based on a state transition probability graph. However, it has a limitation in that the detection speed becomes slower as the number of states increases.

### 3.3. Deep Learning-Based

Recently deep networks have been applied to time series anomaly detection because of their powerful representation learning capabilities [3,4,5,26,27,28,29,30,31,32,33,34]. Deep learning-based approaches can handle the huge feature space of multidimensional time series with less domain knowledge. In recent years, many deep-learning approaches have been developed to detect time series anomalies. To capture the underlying temporal dependencies of time series, a common approach is to use recurrent neural networks, and Du [3] adapted long short-term memory (LSTM) to model time series. However, it cannot be effectively parallelized, making training time-consuming. Shen [4] adopted the dilated recurrent neural network (RNN) to effectively alleviate this problem. The dilated RNN can implement hierarchical learning of dependencies and can implement parallel computing. However, the above approaches all model the time sequence information of time series and pay little attention to the relationship between time series dimensions. Zhang [30] considered this problem and proposed the use of LSTM to model the sequential information of time series while using a one-dimensional convolution to model the relationships between time series dimensions. However, they separately model the relationship between the time sequence information and sequence dimensions of the time series, and this method cannot achieve parallel computing. Recently, deep generative models have also been proposed for anomaly detection. Li [31] proposed MAD-GAN, a variant of generative adversarial networks (GAN), in which they modeled time series using a long short-term memory recurrent neural network (LSTM-RNN) as the generator and discriminator of the GAN. In addition, Audibert et al. [5] also adopted the idea of GAN and proposed USAD; they used the autoencoder as the generator and discriminator of the GAN and used adversarial training to learn the sequential information of time series. However, in practice, it is usually difficult to achieve convergence during GAN training, and it has instability.

## 4. Problem Formulation

An industrial control system measurement device set $E=\left\{E_{1},E_{2},\ldots ,E_{m}\right\}$ contains m measuring devices (sensors and actuators), where $E_{m}$ is the mth device. A multivariate time series $X=\left[x_{1},x_{2},\ldots ,x_{m}\right]\in R^{m\times l}$ is represented as an ordered sequence of *m* dimensions, where *l* is the length of the time series, and *m* is the number of measuring devices. A sequence $X_{t,l}\in R^{m\times l}$ is an overlapping subsequence of a length *l* in the sequence *X* starting at timestamp *t*. We define the set of all overlapping subsequences in a given time series *X*: $X^{{}^{\prime }}=\left\{X_{t,l}|\forall t.0\leq t\leq |X|-l+1\right\}$, where |X| is the length of the series *X*.

Given a set of all subsequences $X^{{}^{\prime }}=\left\{X_{0,l},\ldots ,X_{|X|-l+1,l}\right\}$ of a data series *X*, where |X|−l+1 is the number of all subsequences, and the corresponding label $y=\left\{y_{0},\ldots ,y_{|X|-l+1}\right\}$ represents each time subsequence. In this work, we focus on the time subsequence anomalies. The task of TDRT is to train a model f(x)−>y given an unknown sequence *X* and return *A*, a set of abnormal subsequences.

## 5. Proposed Approach

In this section, we first introduce the overall architecture of our newly proposed TDRT method in Section 5.1. Then, we explain the components of our model in Section 5.2, Section 5.3 and Section 5.4. Finally, we describe the dynamic window selection method in Section 5.5.

### 5.1. Overview

Figure 2 shows the overall architecture of our proposed model. The input to our model is a set of multivariate time series. To model the relationship between temporal and multivariate dimensions, we propose a method to map multivariate time series into a three-dimensional space. Taking the multivariate time series in the bsize time window in Figure 2 as an example, we move the time series by *d* steps each time to obtain a subsequence and finally obtain a group of subsequences in the bsize time window. A detailed description of the method for mapping time series to three-dimensional spaces can be found in Section 5.2.

After completing the three-dimensional mapping, a low-dimensional time series embedding is learned in the convolutional unit. Specifically, we apply four stacked three-dimensional convolutional layers to model the relationships between the sequential information of a time series and the time series dimensions. A detailed description of the low-dimensional embedding learning method can be found in Section 5.3. After learning the low-dimensional embeddings, we use the embeddings of the training samples as the input to the attention learning module. Specifically, we group the low-dimensional embeddings, and each group of low-dimensional embeddings is vectorized as an input to the attention learning module. A detailed description of the attention learning method can be found in Section 5.4.

Furthermore, we propose a method to dynamically choose the temporal window size. Specifically, the dynamic window selection method utilizes similarity to group multivariate time series, and a batch of time series with high similarity is divided into a group. Details of the dynamic window selection method can be found in Section 5.2.

### 5.2. Three-Dimensional Mapping

We first describe the method for projecting a data sequence into a three-dimensional space. Our TDRT method aims to learn relationships between sensors from two perspectives, on the one hand learning the sequential information of the time series and, on the other hand, learning the relationships between the time series dimensions. This facilitates the consideration of both temporal and spatial relationships. In the specific case of a data series, the length of the data series changes over time. To facilitate the analysis of a time series, we define a time window bsize. The length of the time window bsize is *b*. To describe the subsequences, we define a subsequence window sw. The subsequence window sw length is a fixed value *l*. The subsequence window sw is moved by d(d<l) steps each time. Given a time window bsize, the set of subsequences within the time window bsize can be represented as $S=\left\{S_{t},\ldots ,S_{t+b-l+1}\right\}$, where *t* represents the start time of the time window bsize. We reshape each subsequence within the time window bsize into an M×M matrix, where $M=\left\lceil \sqrt{l\times m}\right\rceil$, $\left\lceil .\right\rceil$ represents the smallest integer greater than or equal to the given input. When the value of l×m is less than M×M, add zero padding at the end. Given three adjacent subsequences, we stack the reshaped three matrices together to obtain a three-dimensional matrix. Specifically, the input of the three-dimensional mapping component is a time series *X*, each time window bsize of the time series is represented as a three-dimensional matrix, and the output is a three-dimensional matrix group $S_{M}=\left\{S_{1},\ldots ,S_{|x|-b+1}\right\}$.

To better understand the process of three-dimensional mapping, we have visualized the process. Figure 3 shows a visual representation of a multidimensional time series. Given an M×M matrix, the value of each element in the matrix is between [0,255], where [0,255] corresponds to 256 grayscales. Each matrix forms a grayscale image. We stack three adjacent grayscale images together to form a color image. The three-dimensional representation of time series allows us to model both the sequential information of time series and the relationships of the time series dimensions.

### 5.3. Time Series Embedding

In this work, we focus on subsequence anomalies of multivariate time series. The key is to extract the sequential information and the information between the time series dimensions. The multivariate time series embedding is for learning the embedding information of multivariate time series through convolutional units. The convolution unit is composed of four cascaded three-dimensional residual blocks. The reason we chose a three-dimensional convolutional neural network is that its convolution kernel is a cube, which can perform convolution operations in three dimensions at the same time. Figure 4 shows the embedding process of time series. The time series embedding component learns low-dimensional embeddings for all subsequences of each time window through a convolutional unit. The residual blocks that make up the convolution unit are composed of three-dimensional convolution layers, batch normalization, and ReLU activation functions. We set the kernel of the convolutional layer to 3×3×3 and the size of the filter to 128. The channel size for batch normalization is set to 128. Specifically, the input of the time series embedding component is a three-dimensional matrix group $s_{M}$, which is processed by the three-dimensional convolution layer, batch normalization, and ReLU activation function, and the result of the residual module is the output. After the above steps are carried out many times, the output is $X_{Feature}\in R^{f\times c\times c}$, where *f* is the filter size of the last convolutional layer, and *c* is the output dimension of the convolution operation.

### 5.4. Attention Learning

At the core of attention learning is a transformer encoder. The reason for this design choice is to avoid overfitting of datasets with small data sizes. Figure 5 shows the attention learning method. As described in Section 5.3, the time series encoding component obtains the output feature tensor as $X_{Feature}\in R^{f\times c\times c}$. The feature tensor $X_{Feature}$ is first divided into $\sqrt{n}\times \sqrt{n}$ groups: $F=\left[f_{1},\ldots ,f_{n}\right]$ and then linearly projected to obtain the $V=\left[v_{1},\ldots ,v_{n}\right]$ vector. The linear projection is shown in Formula (Equation 1):(1)$$v_{i}=wf_{i}+b$$
where *w* and *b* are learnable parameters.

Since there is a positional dependency between the groups of the feature tensor, in order to make the position information of the feature tensor clearer, we add an index vector $I=\left[i_{1},\ldots ,i_{n}\right]$ to the vector *V*: $X_{v}=V+I$. Let $X_{v}=\left[x_{1},\ldots ,x_{n}\right]$ be the input for the transformer encoder. We denote the number of encoder layers by *L*. During implementation, the number of encoder layers *L* is set to 6. The transformer encoder is composed of two sub-layers, a multi-head attention layer, and a feed-forward neural network layer. The multi-layer attention mechanism does not encode local information but calculates different weights on the input data to grasp the global information. Given *n* input information $X_{v}=\left[x_{1},\ldots ,x_{n}\right]$, the query vector sequence *Q*, the key vector sequence *K*, and the value vector sequence *V* are obtained through the linear projection of $X_{v}$. The output of each self-attention layer is $H=Vsoftmax(\frac{k^{T}Q}{\sqrt{d_{k}}})$. The output of the multi-head attention layer is concatenated by the output of each layer of self-attention, and each layer has independent parameters. The second sub-layer of the encoder is a feed-forward neural network layer, which performs two linear projections and a ReLU activation operation on each input vector. Residual networks are used for each sub-layer: $LayerNorm(X_{v}+sublayer\left(X_{v}\right))$. The output of the L-layer encoder is fed to the linear layer, and the output layer is a softmax. The loss function adopts the cross entropy loss function, and the training of our model can be optimized by gradient descent methods. The values of the parameters in the network are represented in Table 1.

### 5.5. Dynamic Window Selection

The previous industrial control time series processing approaches operate on a fixed-size sliding window. We now describe how to design dynamic time windows. To describe the correlation calculation method, we redefine a time series $T=\left[t_{1},\ldots ,t_{l}\right]\in R^{m\times l}$, where $t_{i}=\left[t_{i}^{1},\ldots ,t_{i}^{m}\right]\left(1\leq i\leq l\right)$ is an *m*-dimension vector. For instance, when six sensors collect six pieces of data at time *i*, $t_{i}$ can be represented as a vector with the dimension m=6. First, we normalize the time series *T*. The normalization method is shown in Equation (Equation 2).
(2)$$t_{i}^{*}=\frac{t_{i}-{\bar{t}}_{i}}{\sigma }$$
where ${\bar{t}}_{i}$ is the mean of $t_{i}$, and σ is the standard deviation of $t_{i}$. Given a time series *T*, $T^{*}=\left[t_{1}^{*},\ldots ,t_{l}^{*}\right]$ represents the normalized time series, where $t_{i}^{*}=\left[t_{i}^{1*},\ldots ,t_{i}^{m*}\right]$ represents a normalized *m*-dimension vector.

For the time series $X=\left[x_{1},x_{2},\ldots ,x_{m}\right]\in R^{m\times l}$, we define a time window bsize, the size of bsize is not fixed, and there is a set of non-overlapping subsequences $S=\left\{s_{1},s_{2},s_{3}\right\}$ in each time window bsize. The length of each subsequence is determined by the correlation. Figure 6 shows the calculation process of the dynamic window. Given a sequence $t_{i}^{*}$, we calculate the similarity between $t_{i}^{*}$ and $t_{i+1}^{*}$. If the similarity exceeds the threshold τ, it means that $t_{i}^{*}$ and $t_{i+1}^{*}$ are strongly correlated. In this paper, we set τ=90. We group a set of consecutive sequences with a strong correlation into a subsequence. Specifically, when *k* sequences from $t_{i}^{*}$ to $t_{i+1}^{*}$ have strong correlations, then the length of a subsequence of the time window bsize is *k*, that is, $l_{1}=k$. The time window bsize is shifted by the length of one subsequence at a time. In this example, $bsize_{2}$ is moved by $l_{1}$ steps. The correlation calculation is shown in Equation (Equation 3).
(3)$$Cov\left(t_{i}^{*},t_{i+1}^{*}\right)=\frac{\sum_{j=1}^{m}\left(t_{i}^{j*}-{\bar{t}}_{i}^{*}\right)\left(t_{i+1}^{j*}-{\bar{t}}_{i+1}^{*}\right)}{m}$$
where ${\bar{t}}_{i}^{*}$ is the mean of $t_{i}^{*}$, and ${\bar{t}}_{i+1}^{*}$ is the mean of $t_{i+1}^{*}$.

A given time series $X=\left[x_{1},x_{2},\ldots ,x_{m}\right]\in R^{m\times l}$ is grouped according to the correlation to obtain a sub-sequence set $X^{{}^{\prime }}=\left\{X_{1},X_{2},\ldots ,X_{last}\right\}$. The length of all subsequences can be denoted as $L=\left\{l_{1},l_{2},\ldots ,l_{last}\right\}$. In three-dimensional mapping, since the length of each subsequence is different, we choose the maximum length $L_{max}$ of *L* to calculate the value of *M* in order to provide a unified standard. It is worth mentioning that the value of $L_{max}$ is obtained from training and applied to anomaly detection. We reshape each subsequence within the time window bsize into an M×M matrix, $M=\left\lceil \sqrt{L_{max}\times m}\right\rceil$, $\left\lceil .\right\rceil$ represents the smallest integer greater than or equal to the given input. When the value of $l_{i}\left(1\leq i\leq last\right)\times m$ is less than M×M, add zero padding at the end. The rest of the steps are the same as the fixed window method.

## 6. Experimental Setup

This section describes the three publicly available datasets and metrics for evaluation. We evaluate the performance of TDRT and compare it with other state-of-the-art methods.

### 6.1. Public Datasets

Three publicly available datasets are used in our experiments: two real-world datasets, SWaT (Secure Water Treatment) and WADI (Water Distribution), and a simulated dataset, BATADAL (Battle of Attack Detection Algorithms). The characteristics of the three datasets are summarized in Table 2, and more details are described below. When dividing the dataset, the WADI dataset has fewer instances of the test set compared to the SWaT and BATADAL datasets. The reason for this is that the number of instances in the WADI data set has reached the million level, and it is enough to use hundreds of thousands of data instances for testing; more data can be used for training.

SWaT Dataset: SWaT is a testbed for the production of filtered water, which is a scaled-down version of a real water treatment plant. The SWaT testbed is under normal operation for 7 days and under the attack scenario for 4 days. The SWaT dataset is collected for 11 days of data.

WADI Dataset: WADI is an extension of SWaT, and it forms a complete and realistic water treatment, storage, and distribution network. The WADI testbed is under normal operation for 14 days and under the attack scenario for 2 days. The WADI dataset is collected for 16 days of data.

BATADAL Dataset: BATADAL is a competition to detect cyber attacks on water distribution systems. The BATADAL dataset collects one year of normal data and six months of attack data, and the BATADAL dataset is generated by simulation.

### 6.2. Evaluation Metrics

We consider that once there is an abnormal point in the time window bsize, the time window bsize is marked as an anomalous sequence. We adopt Precision (Pre), Recall (Rec), and F1 score (F1) to evaluate the performance of our approach:(4)$$Pre=\frac{TP}{TP+FP}$$
(5)$$Rec=\frac{TP}{TP+FN}$$
(6)$$F1=2\times \frac{Pre\times Rec}{Pre+Rec}$$
where TP represents the true positives, FP represents the false positives, and FN represents the false negatives.

In addition, we use the $F1^{*}$ score to evaluate the average performance of all baseline methods:(7)$$F1^{*}=2\times \frac{\bar{Pre}\times \bar{Rec}}{\bar{Pre}+\bar{Rec}}$$
where $\bar{Pre}$ and $\bar{Rec}$, respectively, represent the average precision and the average recall.

## 7. Experiments and Results

We study the performance of TDRT by comparing it to other state-of-the-art methods (Section 7.1), analyzing the influence of different parameters on the method (Section 7.2), and assessing the performance of the TDRT variant (Section 7.3) through an ablation study (Section 7.4).

### 7.1. Overall Performance

For a comparison of the anomaly detection performance of TDRT, we select several state-of-the-art methods for multivariate time series anomaly detection as baselines.

UAE Frequency: UAE Frequency [35] is a lightweight anomaly detection algorithm that uses undercomplete autoencoders and a frequency domain analysis to detect anomalies in multivariate time series data.MAD-GAN: MAD-GAN [31] is a GAN-based anomaly detection algorithm that uses LSTM-RNN as the generator and discriminator of GAN to focus on temporal–spatial dependencies.OmniAnomaly: OmniAnomaly [17] is a stochastic recurrent neural network for multivariate time series anomaly detection that learns the distribution of the latent space using techniques such as stochastic variable connection and planar normalizing flow.USAD: USAD [5] is an anomaly detection algorithm for multivariate time series that is adversarially trained using two autoencoders to amplify anomalous reconstruction errors.NSIBF: NSIBF [36] is a time series anomaly detection algorithm called neural system identification and Bayesian filtering. It combines neural networks with traditional CPS state estimation methods for anomaly detection by estimating the likelihood of observed sensor measurements over time.

On average, TDRT is the best performing method on all datasets, with an F1 score of over 98%. Table 3 shows the results of all methods in SWaT, WADI, and BATADAL. Table 4 shows the average performance over all datasets. Overall, MAD-GAN presents the lowest performance. This is a GAN-based anomaly detection method that exhibits instability during training and cannot be improved even with a longer training time. The other baseline methods compared in this paper all use the observed temporal information for modeling and rarely consider the information between the time series dimensions. For multivariate time series, temporal information and information between the sequence dimensions are equally important because the observations are related in both the time and space dimensions. In TDRT, the input is a series of observations containing information that preserves temporal and spatial relationships. The performance of TDRT in BATADAL is relatively low, which can be explained by the size of the training set. SWaT and WADI have larger datasets; their training datasets are 56 and 119 times larger than BATADAL, respectively, so the performance on these two datasets is higher than that on the BATADAL dataset.

### 7.2. Effect of Parameters

Different time windows have different effects on the performance of TDRT. In this section, we study the effect of the parameter bsize on the performance of TDRT. Considering that bsize may have different effects on different datasets, we set different time windows on the three datasets to explore the impact of time windows on performance. The size of the time window bsize can have an impact on the accuracy and speed of detection. Since different time series have different characteristics, an inappropriate time window may reduce the accuracy of the model. In addition, it is empirically known that larger time windows require waiting for more observations, so longer detection times are required. Essentially, the size of the time window is reflected in the subsequence window sw. Therefore, we take sw as the research objective to explore the effect of time windows on model performance. Figure 7 shows the results on three datasets for five different window sizes. When the subsequence window sw is set to 10, TDRT shows the best performance on the SWaT dataset. The performance of TDRT on the WADI dataset is relatively insensitive to the subsequence window, and the performance on different windows is relatively stable. Considering that a larger subsequence window requires a longer detection time, we set the subsequence window sw of the WADI dataset to five. The performance of TDRT on the BATADAL dataset is relatively sensitive to the subsequence window. When the subsequence window sw=1, TDRT shows the best performance on the BATADAL dataset.

### 7.3. Performance of TDRT-Variant

The average F1 score for the TDRT variant is over 95%. Recall that we studied the effect of different time windows on the performance of TDRT. Choosing an appropriate time window is computationally intensive, so we propose a variant of TDRT that provides a unified approach that does not require much computation. In this experiment, we investigate the effectiveness of the TDRT variant. The results are shown in Figure 8. As can be seen, the proposed TDRT variant, although relatively less effective than the method with carefully chosen time windows, outperforms other state-of-the-art methods in the average F1 score.

### 7.4. Ablation Study

The average F1 score improved by 5.6% relative to methods that did not use attentional learning. Using the SWaT, WADI, and BATADAL datasets, we investigate the effect of attentional learning. Figure 9 shows a performance comparison in terms of the F1 score for TDRT with and without attention learning. The ablated version of TDRT has a lower F1 score than that of TDRT without ablation. In conclusion, ablation leads to performance degradation.

## 8. Conclusions

In this paper, we make the following two key contributions: First, we propose TDRT, an anomaly detection method for multivariate time series, which simultaneously models the order information of multivariate time series and the relationships between the time series dimensions. By extracting spatiotemporal dependencies in multivariate time series of Industrial Control Networks, TDRT can accurately detect anomalies from multivariate time series. In comprehensive experiments on three high-dimensional datasets, TDRT outperforms state-of-the-art multivariate time series anomaly detection methods. Our results show that the average F1 score of TDRT is over 98%. Second, we propose a method to automatically select the temporal window size called the TDRT variant. In comprehensive experiments on three high-dimensional datasets, the TDRT variant provides significant performance advantages over state-of-the-art multivariate time series anomaly detection methods. Our results show that the average F1 score of the TDRT variant is over 95%.

A limitation of this study is that the application scenarios of the multivariate time series used in the experiments are relatively homogeneous. In the future, we will conduct further research using datasets from various domains, such as natural gas transportation and the smart grid.

## Figures and Tables

**Figure 1 entropy-25-00180-f001:**
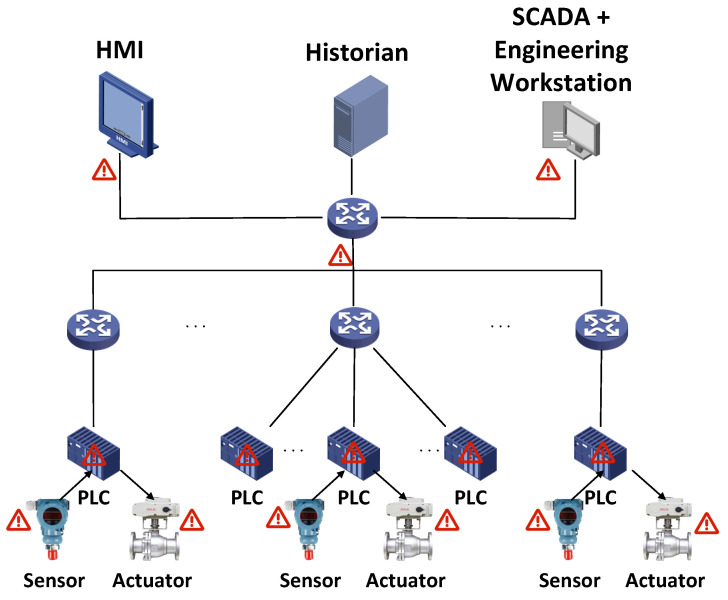
ICS architecture and possible attacks.

**Figure 2 entropy-25-00180-f002:**
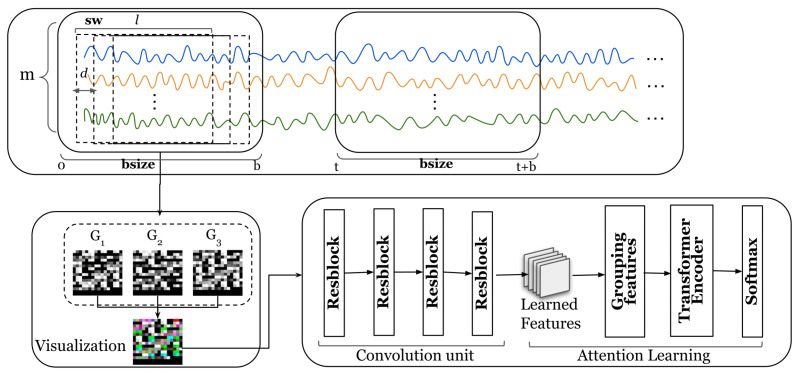
Overall architecture of the TDRT model. The m lines of different colors represent the time series collected by m sensors.

**Figure 3 entropy-25-00180-f003:**
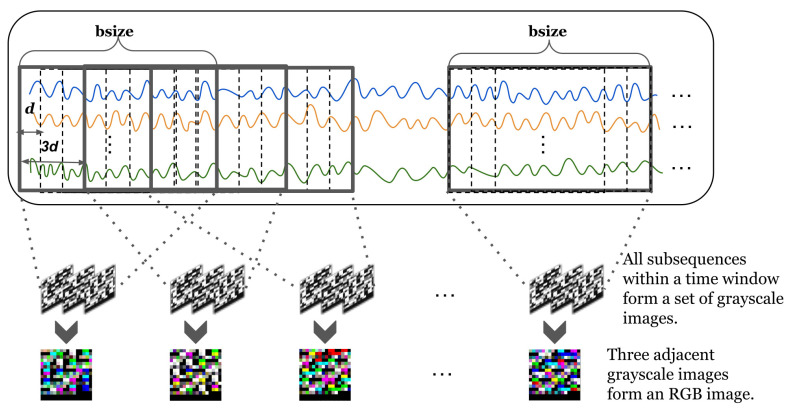
Visual representation of a multidimensional time series. Lines of different colors represent different time series.

**Figure 4 entropy-25-00180-f004:**
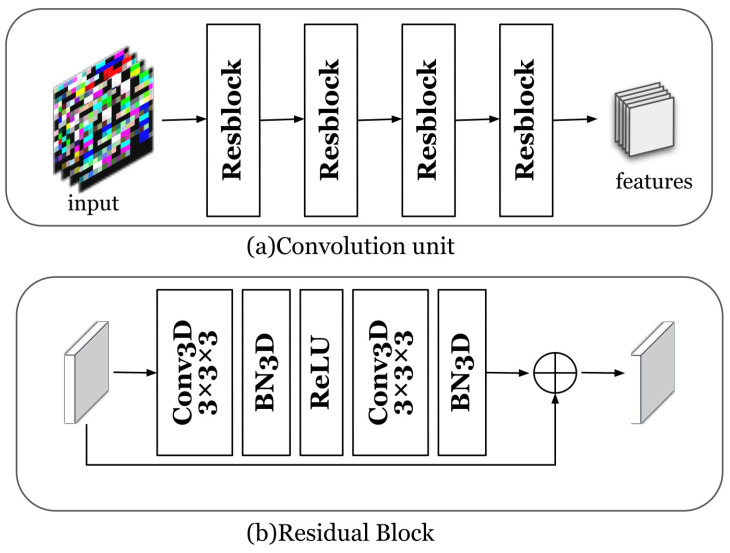
Time series embedding: (**a**) the convolution unit; (**b**) the residual block component.

**Figure 5 entropy-25-00180-f005:**
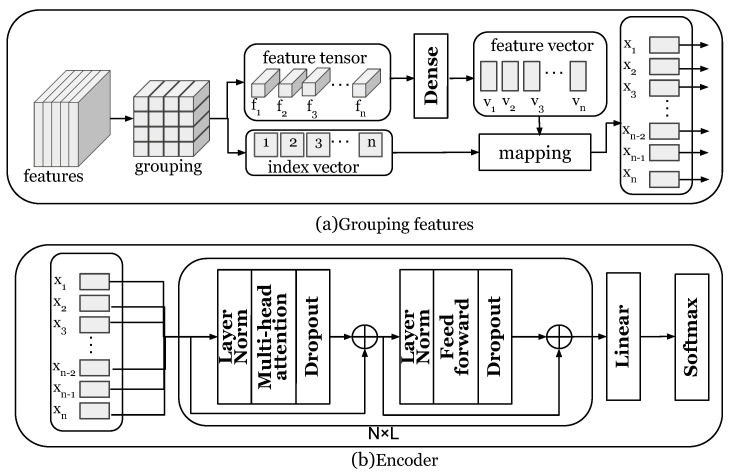
Attention learning.

**Figure 6 entropy-25-00180-f006:**
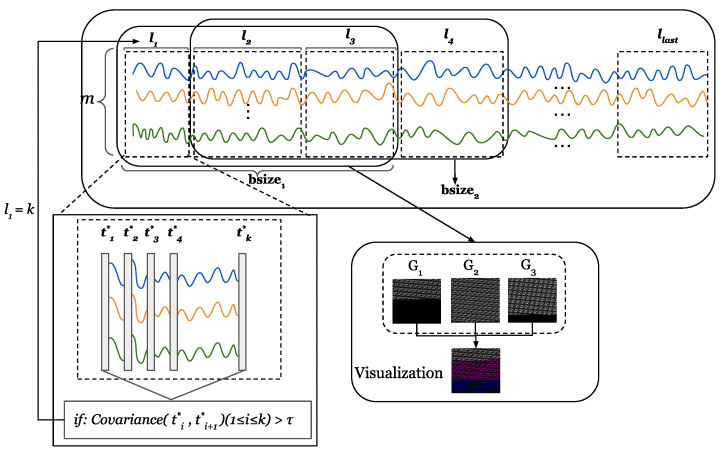
The process of dynamic window calculation.The m lines of different colors represent the time series collected by m sensors.

**Figure 7 entropy-25-00180-f007:**
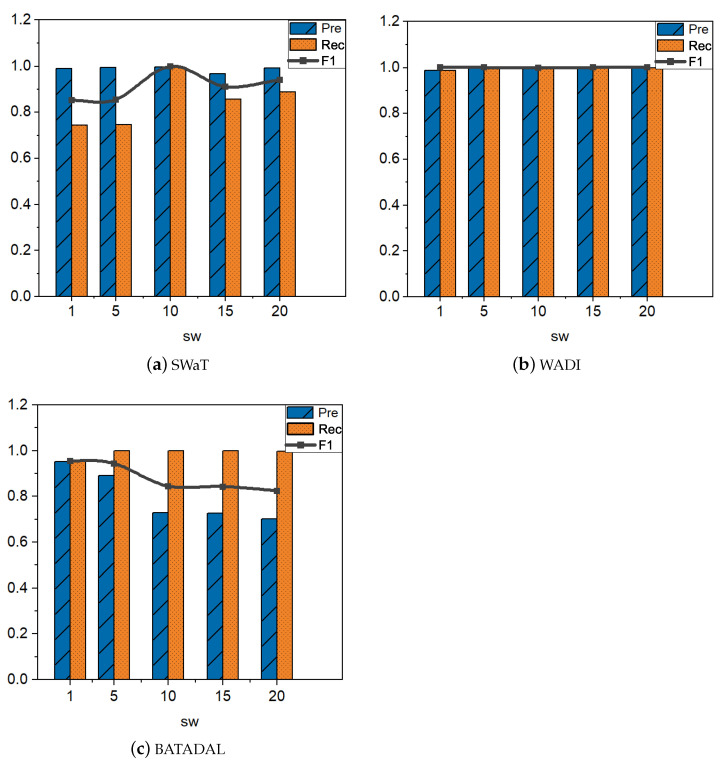
The effect of the subsequence window sw on Precision, Recall, and F1 score.

**Figure 8 entropy-25-00180-f008:**
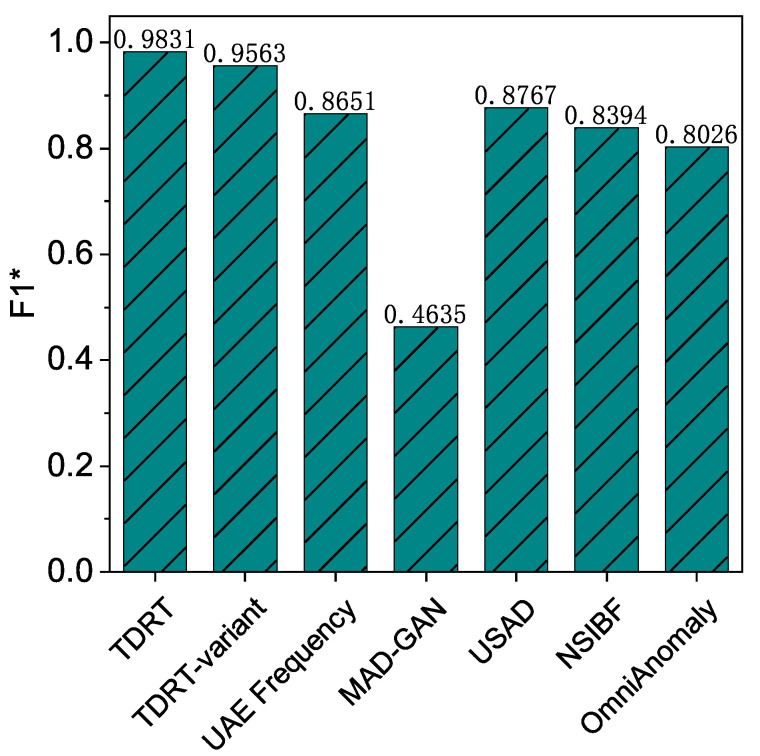
Performance of all solutions.

**Figure 9 entropy-25-00180-f009:**
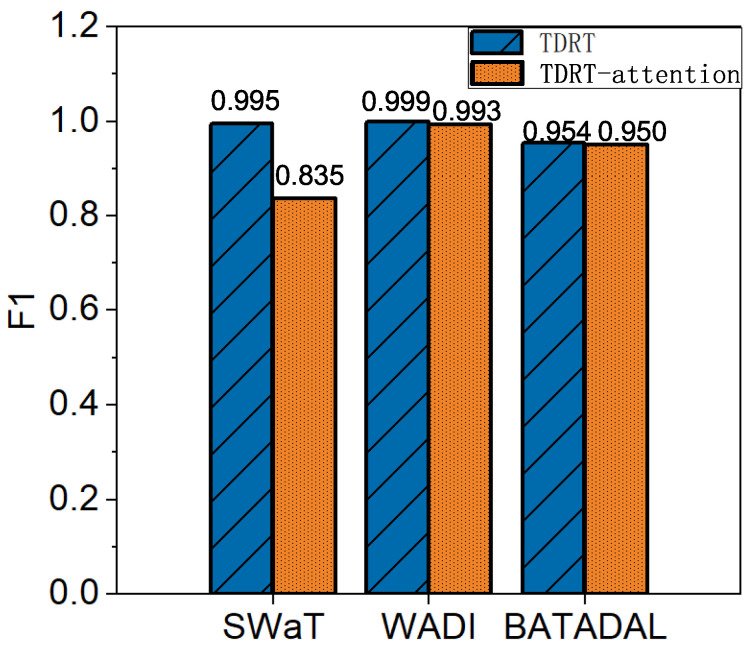
Impact with and without attention learning on TDRT.

**Table 1 entropy-25-00180-t001:** Parameter settings.

Parameter	Value
Epoch	100
Learning rate	0.001
L	6
Dropout	0.1
Heads	4

**Table 2 entropy-25-00180-t002:** Details of the three datasets.

Dataset	Features	Train	Test
SWaT	51	496,802	449,920
WADI	123	1,048,571	172,801
BATADAL	43	8761	4177

**Table 3 entropy-25-00180-t003:** Precision (Pre), recall (Rec), and F1 score results (as %) on various datasets.

Methods	SWaT			WADI			BATADAL		
	Pre	Rec	F1	Pre	Rec	F1	Pre	Rec	F1
UAE Frequency	0.9110	0.8600	0.8850	0.9160	0.6400	0.7540	0.9180	0.9610	0.9390
MAD-GAN	0.9897	0.6374	0.7700	0.4144	0.3392	0.3700	0.0631	0.3450	0.0953
USAD	0.9851	0.6618	0.7919	0.9870	0.7400	0.8460	0.9342	1	0.9660
NSIBF	0.9820	0.8630	0.9190	0.9150	0.8870	0.9010	0.7143	0.6818	0.6977
OmniAnomaly	0.9790	0.7570	0.8540	0.8460	0.8930	0.8690	0.8712	0.5251	0.6250
TDRT-V	0.9100	0.9476	0.9284	0.9865	1	0.9932	0.8979	1	0.9462
TDRT	0.9971	0.9947	0.9959	0.9998	1.0	0.9999	0.9517	0.9555	0.9536

**Table 4 entropy-25-00180-t004:** Average performance (±standard deviation) over all datasets.

	Pre	Rec	F1	F1*
UAE Frequency	0.9150	0.8203	0.8593	0.8651
MAD-GAN	0.4891	0.4405	0.4118	0.4635
USAD	0.9688	0.8006	0.8679	0.8767
NSIBF	0.8704	0.8106	0.8392	0.8394
OmniAnomaly	0.8987	0.7250	0.7827	0.8026
TDRT-V	0.9314	0.9825	0.9559	0.9563
TDRT	0.9828	0.9834	0.9831	0.9831

## Data Availability

Not applicable.

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
