# Peer review of "A Three-Dimensional ResNet and Transformer-Based Approach to Anomaly Detection in Multivariate Temporal–Spatial Data"

_entropy, 2023, doi:10.3390/e25020180_

Round 1

Reviewer 1 Report

Dear authors, thank you for the opportunity to read such a well written paper. The addressed problem and the solution are clearly described as well as the results which represent an improvement over the state-of-the-art. There are some minor details that you might need to address:

1. The first sentence in the abstract is superfluous. You might want to consider rephrasing by merging it with the next sentence or removing it altogether.

2. Row 14 – video instead of vedio

3. The example on line 56 should contain more detail. It is not clear from the formulation what the issue and the expected behavior are (and why).

4. At line 268, it probably should be X_Feature

5. When describing the parameters, you wrote “The length of the time window bsize is b. To describe subsequences, we define a subsequence window sw.” However, in Figure 7 which should be “The effect of the time window bsize on Precision, Recall and F1 score”, the description of the X axis reads “sw” instead of “bsize”.

6.Figure 8 caption is misleading, it should the F1 score for all solutions, not only the TDRT-variant.

Thank you and good luck!

Reviewer 2 Report

I appreciate your work.

I have a question: why the examples of test set for the WADI dataset is so low compared to the training and also the divisions of the other datasets? you should provide a motivation in the text.

Reviewer 3 Report

Please add more description in the section of "Conclusion".
